# Effect of Different Plant Communities on Fine Particle Removal in an Urban Road Greenbelt and Its Key Factors in Nanjing, China

Congzhe Liu [1,2,3], Anqi Dai [1], Yaou Ji [1], Qianqian Sheng [1,2,3,4,*] and Zunling Zhu [1,2,3,4,5,*]

1   College of Landscape Architecture, Nanjing Forestry University, Nanjing 210037, China
2   Jin Pu Research Institute, Nanjing Forestry University, Nanjing 210037, China
3   Research Center for Digital Innovation Design, Nanjing Forestry University, Nanjing 210037, China
4   Co-Innovation Center for Sustainable Forestry in Southern China, Nanjing Forestry University, Nanjing 210037, China
5   College of Art & Design, Nanjing Forestry University, Nanjing 210037, China
*   Correspondence: qqs@njfu.edu.cn (Q.S.); zhuzunling@njfu.edu.cn (Z.Z.)

**Abstract:** Determining the relationships between the structure and species of plant communities and their impact on ambient particulate matter (PM) is an important topic in city road greenbelt planning and design. The correlation between the distribution of plant communities and ambient PM concentrations in a city road greenbelt has specific spatial patterns. In this study, we selected 14 plant-community-monitoring sites on seven roads in Nanjing as research targets and monitored these roads in January 2022 for various parameters such as PM with aerodynamic diameters $\leq 10$ μm ($PM_{10}$) and PM with aerodynamic diameters $\leq 2.5$ μm ($PM_{2.5}$). We used a spatial model to analyze the relationship between the concentrations of ambient $PM_{10}$ and $PM_{2.5}$ and the spatial heterogeneity of plant communities. The consequences revealed that the composition and species of plant communities directly affected the concentrations of ambient PM. However, upon comparing the PM concentration patterns in the green community on the urban road, we found that the ability of the plant community structures to reduce ambient PM is in the order: trees + shrubs + grasses > trees + shrubs > trees + grasses > pure trees. Regarding the reduction in ambient PM by tree species in the plant community (conifer trees > deciduous trees > evergreen broad-leaved trees) and the result of the mixed forest abatement rate, coniferous + broad-leaved trees in mixed forests have the best reduction ability. The rates of reduction in $PM_{10}$ and $PM_{2.5}$ were 14.29% and 22.39%, respectively. We also found that the environmental climate indices of the road community, temperature, and traffic flow were positively correlated with ambient PM, but relative humidity was negatively correlated with ambient PM. Among them, $PM_{2.5}$ and $PM_{10}$ were significantly related to temperature and humidity, and the more open the green space on the road, the higher the correlation degree. $PM_{10}$ is also related to light and atmospheric radiation. These characteristics of plant communities and the meteorological factors on urban roads are the foundation of urban greenery ecological services, and our research showed that the adjustment of plant communities could improve greenbelt ecological services by reducing the concentration of ambient PM.

**Keywords:** plant community; particulate matter; spatial distribution; urban greenbelt

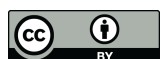

## 1. Introduction

With the development of industrialization and urbanization, air pollution has become an environmental problem that must be solved [1]. The concentration of atmospheric pollutants in many cities around the world exceeds standards, and these unhealthy conditions are getting worse [2]. To standardize the control of particulate matter (PM) pollution concentrations, the "National Air Pollution Prevention and Control Law" was revised in 2015 in China [3]. In addition to promoting and using renewable energy and reducing motor

vehicle emissions, programs that can alleviate road traffic pollution are being increasingly studied [4].

Roadside greenbelts (areas of open land on which building is restricted) can improve air quality by intercepting and fixing ambient dirt to block dust [5]. At present, studies on solving environmental pollution problems should include more than the supervision of pollution sources. Urban road greenbelts are considered essential quantitative indicators for evaluating the environmental benefits of green spaces [6]. Previous research showed that the concentration of PM with aerodynamic diameters $\leq 2.5$ μm ($PM_{2.5}$) at each research point presented a significant positive correlation with the relative humidity in the community. However, the correlation was insignificant regarding canopy closure, lawn coverage, and atmospheric pressure [7]. Plants can shield 36% of the particulate matter, and different plant communities can reduce the concentration of $NO_2$ and $SO_2$ by 10% to 30% [8]. In addition, the three-dimensional green mass of different plant communities in open spaces exhibited a significantly negative correlation with pollutants [9]. The effect is also influenced by external environmental factors, such as wind speed (WS) and direction [10]. In plant communities, substantial differences were also found for PM retained by different species, such as arbors, shrubs, herbs, conifers, and broadleaf deciduous and evergreen trees [11].

Research has shown a strong association between exposure to ambient PM and the respiratory system [12]. In China, the proportion of deaths caused by environmental $PM_{2.5}$ pollution is as high as 16.2% of the total number of deaths, which is the fourth-highest major factor contributing to death according to 2014 estimates [13]. Ambient PM in Nanjing occurs from motor vehicles and coal combustion [14]. Owing to the differences in energy consumption and meteorological conditions between different seasons, there are daily and seasonal changes in the ambient PM concentration in Nanjing. Li's study showed that the annual mean mass fractions of $PM_{2.5}$ and $PM_{10}$ in Nanjing exceeded the secondary standard limits of the "Ambient Air Quality Standard" by 44% and 38%, respectively [15]. The concentrations of $PM_{2.5}$ and $PM_{10}$ exhibit drastic seasonal changes. $PM_{2.5}$ and $PM_{10}$ concentrations in spring are 3.1 and 1.9 times lower than in winter, respectively, primarily caused by intensive emissions from coal burning for domestic heating [16]. The daily peak concentrations of PM occur at 7:00–8:00 and 19:00–20:00 [17]. It was proposed that 30% of $PM_{2.5}$ on city roads could be attributed to sources outside Nanjing [18]. Therefore, PM concentrations in Nanjing are strongly influenced by wind direction [19]. The number and type of motor vehicles directly influence gas emissions. The greater the traffic flow in the morning and evening peak hours, the higher the daily pollutant concentration [17].

There are insufficient available data regarding how ambient PM concentrations vary near smaller-scale plant communities (e.g., parks, public greenery, or roadside greenbelt). There is a need to incorporate the reduction in ambient PM in the environment into the planning and development process to maximize its ecological service value. In this study, 14 sample plots on seven roads in Nanjing were researched because of different plant species and various plant community structures. The systematic planning of plant communities could proactively promote the benefits of urban greenbelts on air quality. However, to achieve this, we need a deeper comprehension of the relationships between plant community characteristics and species with microclimates and their effects on the reduction in ambient PM concentrations. Therefore, the purposes of this research were to: (1) record the effects of plant community spatial heterogeneity on environmental ambient $PM_{10}$ and $PM_{2.5}$ concentrations, (2) determine which plant community characteristics had the greatest impact on reductions in $PM_{10}$ and $PM_{2.5}$ levels in a roadside greenbelt, (3) determine the seasonal and diurnal concentration patterns of ambient $PM_{10}$ and $PM_{2.5}$, and (4) explore the influence of environmental factors on the road on the concentration of ambient particulate matter.

## 2. Materials and Methods

### 2.1. Study Area

Nanjing City is in the lower reaches of the Yangtze River, in the eastern part of China, with latitude 31°14′–32°37′ N and longitude 118°22′–119°14′ E. The terrain of Nanjing is long from north to south and narrow from east to west, in a north–south direction; to the south is a geomorphic complex composed of topographic units such as low mountains, hills, valley plains, lakeside plains, and riverside land. The Nanjing climate can be classified as northern subtropical humid, with four distinct seasons and abundant rainfall. Based on the survey of all roads in the Nanjing urban area, we selected seven roads in the five main metropolitan regions of Nanjing (Jianye District, Xuanwu District, Gulou District, Qixia District, and Jiangning District). The roads overlay areas of arbors, shrubs, grasses, etc. The roadside green space with different plant communities is also adjacent to different road grades and directions, as well as the downtown area and the suburbs (Figure 1). The main research area included the road lanes, a cross-section of the greenbelt, and plant communities.

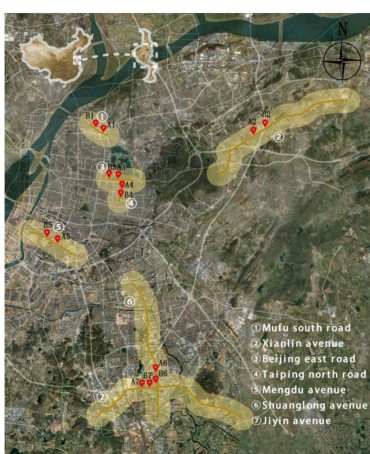

**Figure 1.** Study area (See Table 1 for details), the seven roads in Nanjing, Jiangsu, China. The research roads are marked with yellow lines and the research sites are marked with red points.

### 2.2. Plant Community Description

A total of 14 plant communities distributed along roadsides in Nanjing were selected. The Swedish quadrant approach was used to investigate these plant communities. According to the empirical value of the minimum area of road greenery communities, our selected research sites were square-shaped (20 × 20 m) sample plots. The plant communities and main species are presented in Table 1. We used the dominant species to identify all associations. The combination and species of 14 plant communities (Table 1) were recognized as follows:

1. Coniferous tree communities, including A2 *Cedrus deodara* (Roxb.) G. Don;
2. Evergreen tree communities, including B1 *Malus halliana* + *Cinnamomum camphora* + *Osmanthus fragrans* + *Acer palmatum* and B5 *C. camphora* + *Ginkgo biloba* + *A. palmatum*;
3. Mixed broadleaf and coniferous trees, including B2 *C. deodara* + *C. camphora* and B4 *O. fragrans* + *Phyllostachys edulis* + *Magnolia grandiflora* + *Zelkova serrata* + *Carya illinoinensis* + *Prunus serrulata*;
4. Mixed deciduous and evergreen trees, including A4 *Metasequoia glyptostroboides* + *Carya illinoinensis* + *P. serrulata*;
5. Deciduous trees, including B3 *Styphnolobium japonicum* + *P. serrulata*;
6. Mixed evergreen and deciduous trees, including A5 *C. camphora* + *G. biloba*, A6 *Sapindus saponaria* + *Prunus cerasifera*. "Atropurpurea" + *C. camphora* + *G. biloba*, B6 *P. cerasifera* "Atropurpurea" + *G. biloba* + *Cornus wilsoniana* + *Albizia julibrissin* + *C. camphora*, A7 *Koelreuteria paniculata* + *Photinia serratifolia* + *C. camphora* + *Platanus*

acerifolia, B7 Koelreuteria paniculata + O. fragrans + Lagerstroemia indica + P. serrulata + Nerium oleander + Prunus persica;

7.  Blank control group, including square A3 and lawn A1 Cynodon dactylon.

**Table 1.** Structural characteristics of 14 plant communities in Nanjing road greenbelt.

| Street | Site | Street Tree Greenbelt Species |
|---|---|---|
| Mufu South Road | A1 | Cynodon dactylon |
| | B1 | Malus halliana + Cinnamomum camphora + Osmanthus fragrans + Acer palmatum − Photinia serratifolia + Euonymus japonicus "Aurea-marginatus" + Loropetalum chinense var. rubrum + Ligustrum lucidum − Ophiopogon bodinieri |
| | Elevation (1:300) |  Elevation of research site A1 B1, Mufu South Road |
| Xianlin Avenue | A2 | Cedrus deodara − Pennisetum alopecuroides |
| | B2 | Cedrus deodara + Cinnamomum camphora − Ophiopogon bodinieri |
| | Elevation (1:300) |  Elevation of research site A2 B2, Xianlin Ave. |
| Beijing East Road | A3 | Rosa chinensis − Rhododendron simsii |
| | B3 | Styphnolobium japonicum + Prunus serrulata − Zoysia japonica |
| | Elevation (1:300) |  Elevation of research site A3 B3, Beijing East Road. |

**Table 1.** *Cont.*

| Street | Site | Street Tree Greenbelt Species |
|---|---|---|
| Taiping North Road | A4 | *Metasequoia glyptostroboides + Carya illinoinensis + Prunus serrulata − Photinia serratifolia + Nandina domestica + Mahonia bealei* |
| | B4 | *Osmanthus fragrans + Phyllostachys edulis + Magnolia grandiflora + Zelkova serrata + Carya illinoinensis + Prunus serrulata − Euonymus japonicus* "Aurea-marginatus" *+ Photinia × fraseri + Loropetalum chinense* var. *rubrum − Ophiopogon bodinieri* |
| | Elevation (1:300) |  Elevation of research site A4 B4, Taiping North Road. |
| Mengdu Street | A5 | *Cinnamomum camphora + Ginkgo biloba* |
| | B5 | *Cinnamomum camphora + Ginkgo biloba + Acer palmatum − Rhododendron simsii + Pittosporum tobira + Pittosporum tobira − Ophiopogon japonicus + Ophiopogon bodinieri* |
| | Elevation (1:300) |  Elevation of research site A5 B5, Mengdu Street. |
| Shuanglong Avenue | A6 | *Sapindus saponaria + Prunus cerasifera* f. *atropurpurea + Cinnamomum camphora + Ginkgo biloba − Ophiopogon bodinieri* |
| | B6 | *Prunus cerasifera* f. *atropurpurea + Ginkgo biloba + Cornus wilsoniana + Albizia julibrissin + Cinnamomum camphora − Photinia serratifolia − Ophiopogon bodinieri* |
| | Elevation (1:300) |  Elevation of research site A6 B6, Shuanglong Ave. |

**Table 1.** *Cont.*

| Street | Site | Street Tree Greenbelt Species |
|---|---|---|
| | A7 | *Koelreuteria paniculata + Photinia serratifolia + Cinnamomum camphora + Platanus acerifolia* |
| | B7 | *Koelreuteria paniculata + Osmanthus fragrans + Lagerstroemia indica + Prunus serrulata + Nerium oleander + Prunus persica − Euonymus japonicus "Aurea-marginatus" + Ligustrum × vicaryi − Ophiopogon japonicus* |
| Jiyin Avenue | Elevation (1:300) |  |
| | | Elevation of research site A7 B7, Jiyin Ave. |

### 2.3. Monitoring Network and Sampling

A total of 14 plots distributed within 14 plant communities on seven roads in Nanjing were established (Figure 1). The plots included representatives from plant communities on urban roads in Nanjing. Plots A1 and B1 were located on Mufu South Road of Gulou district; samples A2 and B2 were located on Xianlin Avenue of Jiangning district; samples A3 and B3 were located on Beijing East Road of Xuanwu district; samples A4 and B4 were located on Taiping North Road of Xuanwu district; samples A5 and B5 were located on Mengdu Avenue of Jianye district; samples A6 and B6 were located on Shuanglong Avenue of Jiangning district; and samples A7 and B7 were located on Jiying Avenue of Jiangning district. The two unshaded paved spaces, which are located on Beijing East Road of Xuanwu district (A3), and another located on Mufu South Road of Gulou district (A1), were treated as the control group (CK). We recorded the characteristics for 14 plants communities, as follows: (1) geographical location information, including altitude (A), longitude and latitude (LO and LA), distance from the road (DR), and distance from the edge of buildings (DB); (2) weather factors including temperature (T), relative humidity (RH), WS, the direction of the wind (DW), illumination (I), radiation (R), atmospheric pressure (AP), and noise (N); (3) the composition of each community, including the height of trees (H), diameter at breast height (DBH), and canopy area (CA). The geographic coordinates A, LO and LA, and the weather factors were recorded using a self-developed multi-function lifting environmental detector (Table 2). The H and DBH were measured using an inclinometer and DBH with a ruler, and CA was calculated by the shape of the tree crown. DR and DB were determined using ArcGIS v. 10.0 (ESRI, Redlands, CA, USA).

**Table 2.** Measuring range and accuracy of lifting environmental detector.

| Testing Content | Measurement Range | Resolution Ratio | Precision | Testing Content | Measurement Range | Resolution Ratio | Precision | |
|---|---|---|---|---|---|---|---|---|
| PM$_{2.5}$ | 0–1000 µg/m$^3$ | 1 µg/m$^3$ | ±10 µg/m$^3$ | PM$_{10}$ | 0–1000 µg/m$^3$ | 1 µg/m$^3$ | ±10 µg/m$^3$ | |
| NO$_2$ | 0–100 ppm | 0.01 ppm | ≤±3% | O$_3$ | 0–100 ppm | 0.01 ppm | ≤±3% | |
| Temperature | −40–80 °C | 0.1 °C | ±2 °C | Relative humidity | 0–100%RH | 0.1%RH | ±2%RH | |
| Wind speed | 0–60 m/s | 0.1 m/s | ±0.5 m/s | Atmospheric pressure | 300–1200 hpa | 1 pa | ±1.5 hpa | |
| Radiation | 0–2000 µw/cm$^2$ | 1 | ±1 µw/cm$^2$ | Illumination | 0–300 KLux | 0.1 KLux | ±0.1 KLux |  |
| Noise | 30–120 dB | 0.1 dB | <2% | | | | | |

### 2.4. Data Collection

The measurements of ambient $PM_{10}$ and $PM_{2.5}$ concentrations were determined from the 14 plots on seven roads. Briefly, a 20 × 20 m plot was set by distances with 3–4 monitoring sites in a lateral direction of the road (Figure 2). Measurements were taken five times by lifting poles. Each monitoring site recorded pollutant gas concentrations at heights of 0 m, 0.5 m, 1.5 m, 3 m, and 6 m above the ground (Figure 2). The device had a measurement range of 1–1000 µg/m$^3$, a resolution of 1 µg/m$^3$ on PM, and the error was $\leq \pm$ 10 µg/m$^3$. The working environment had a temperature range from −40 to 80 °C, a resolution of 0.1 °C, and an error of $\leq \pm 2$ °C. The detector had a range of 0–100% RH with a resolution of 0.1% RH and an error of $\leq \pm 2$% RH. Other meteorological indicators are shown in Table 2. Data were collected in January 2022. The concentrations of ambient $PM_{10}$ and $PM_{2.5}$ and weather factors (R, RH, T, AP, N, I, WS, and WD) were collected under conditions with WS < 2 m/s from 7:00 to 19:00 on one day at each of the 14 different communities and were recorded separately at 7:00–9:00, 12:00–14:00, and 17:00–19:00 at each plot. The data were recorded under specific WS conditions so that the effect on the PM measurements would be negligible [20].

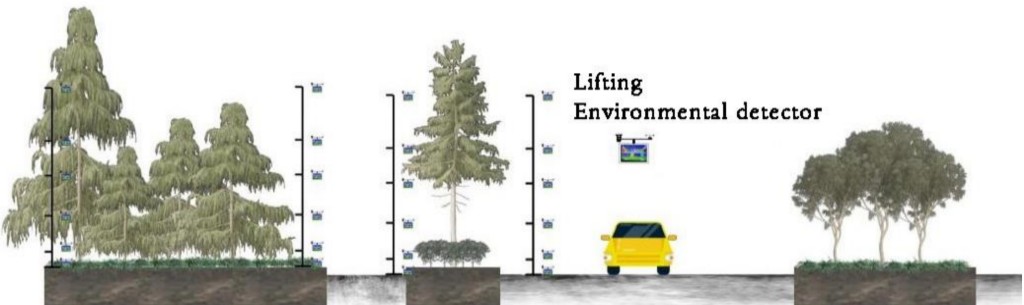

**Figure 2.** Monitoring sites of lifting environmental detector.

### 2.5. Data Analysis

Descriptive statistical analyses were performed to evaluate the average, partial correlation analysis, standard deviation, and correlation for each measured parameter. A partial correlation analysis was performed using the corr package in R (3.6.0) to understand the relationships between the variables and PM. Partial correlation analysis is a method that can eliminate the influence of other elements and calculate the strength and direction of the correlation between two factors alone. The calculation formula is:

$$R_{xy,z} = \frac{R_{xy} - R_{xz} R_{yz}}{\sqrt{\left(1 - R_{xz}^2\right)\left(1 - R_{yz}^2\right)}}, \tag{1}$$

where $R_{xyz}$ is the partial correlation coefficient between x and y after removing the influence of z; $R_{xy}$, $R_{xz}$, and $R_{yz}$ are the correlation coefficients between the two factors.

The formula for calculating the percentage of pollutant purification in green belts with different plants [21]:

$$P_n = (C_c - C_0)/C_c \times 100\%, \tag{2}$$

where $P_n$ is the purification percentage of various pollutants by the greenbelt; $C_c$ is the pollutant concentration on the side of the motor vehicle lane closest to the greenbelt; $C_0$ is the pollutant concentration on the side of the greenbelt farthest from the edge of the motor vehicle lane (control concentration).

## 3. Results

### 3.1. Influence of Environmental Factors on Ambient PM$_{10}$ and PM$_{2.5}$ Concentration

#### 3.1.1. Traffic Flow

The daily variation in the concentrations of ambient PM$_{10}$ and PM$_{2.5}$ on the seven roads in Nanjing increased with the increase in traffic flow and had a significant positive correlation with the traffic flow (Figure 3). Typically, the ambient PM$_{10}$ and PM$_{2.5}$ concentrations in Nanjing are relatively low, essentially in accordance with the national secondary air quality standards, but there are differences from the air quality guidelines. Owing to the minimal difference between morning-peak and off-peak traffic flow, but a large night-peak traffic flow (Table 3), the ambient PM$_{10}$ and PM$_{2.5}$ during the morning peak are far lower than those in the noon off-peak and evening peaks on Xianlin Avenue and Mufu South Road. However, on Mengdu Avenue, Beijing East Road, and Jiyin Avenue, the changes in the ambient particulate matter vary with traffic flow. The concentration of ambient PM$_{10}$ and PM$_{2.5}$ on Taiping North Road and Shuanglong Avenue had no significant relationship with peak hours of traffic flow. We observed that the fluctuation of the concentration of ambient PM$_{10}$ and PM$_{2.5}$ on the road was not only affected by traffic flow but also affected by meteorological factors (Table 3).

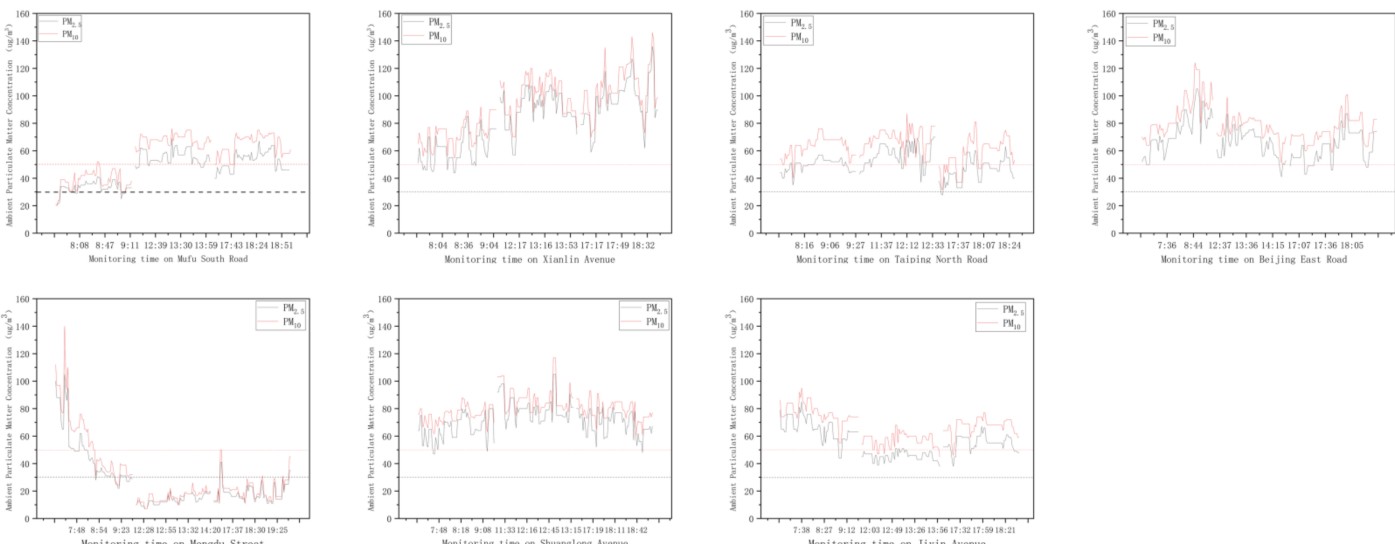

**Figure 3.** Daily dynamic changes in ambient PM$_{10}$ and PM$_{2.5}$ concentrations on seven research roads.

**Table 3.** Daily traffic flow on seven roads in Nanjing.

| Street | Traffic Flow | | |
| --- | --- | --- | --- |
| | Morning | Noon | Evening |
| Mufu South Road | 40/5 min | 41/5 min | 64/5 min |
| Xianlin Avenue | 91/5 min | 69/5 min | 164/5 min |
| Taiping North Road | 120/5 min | 40/5 min | 62/5 min |
| Beijing East Road | 88/5 min | 62/5 min | 89/5 min |
| Mengdu Street | 97/5 min | 100/5 min | 117/5 min |
| Shuanglong Avenue | 206/5 min | 101/5 min | 170/5 min |
| Jiyin Avenue | 140/5 min | 41/5 min | 76/5 min |

#### 3.1.2. Weather Factors

By analyzing the partial correlations of various meteorological factors (including temperature, RH, WS, AP, I, and R) and road pollutants on the seven roads under winter conditions, we found that no uniform correlation existed between meteorological factors and road pollutants. These results indicated that the concentration of road pollutants is af-

fected by complex meteorological factors. Based on the analysis and comparison of different roads, the correlation between $PM_{10}$, $PM_{2.5}$, and meteorological factors is significant.

Temperature and humidity have a significant correlation with $PM_{2.5}$ and $PM_{10}$, because temperature and humidity have a direct relationship with plant fine particle removal; this correlation is higher the more open the green space is on the road. The correlation between wind speed and fine particles is the lowest. Because the wind speed was lower than 2 m/s during experimental monitoring, the impact of wind speed on pollutants can be ignored in this study. The atmospheric pressure is mainly related to $PM_{2.5}$, but not to $PM_{10}$, which indicates that atmospheric pressure only affects particles with smaller diameters. On the contrary, light and atmospheric radiation have a high correlation with $PM_{10}$ but have a low impact on $PM_{2.5}$. The relationship between noise and fine particles is relatively high on Jiyin Avenue, which may be related to the reflection of sound from the height of the street canyon, considering the height and canopy density of the trees (Table 4).

**Table 4.** Partial correlation analysis of inhalable particulate matter and meteorological factors.

| Road | Pollutant | Temperature/$°C$ | Relative Humidity | Wind Speed/m/s | Atmospheric Pressure/pa | Atmospheric Radiation/uw/cm$^2$ | Light Radiation/Lux | Noise/DB |
|---|---|---|---|---|---|---|---|---|
| Beijing East Road | $PM_{2.5}$ | 0.365 * | 0.300 | 0.214 | −0.106 | 0.028 | −0.027 | 0.034 |
| | $PM_{10}$ | −0.405 * | −0.319 * | −0.145 | 0.203 | −0.083 | 0.095 | −0.044 |
| Mufu South Road | $PM_{2.5}$ | 0.316 * | −0.012 | 0.011 | 0.365 ** | −0.261 | 0.355 * | −0.130 |
| | $PM_{10}$ | −0.432 ** | 0.001 | 0.147 | −0.322 | 0.253 | −0.312 * | 0.173 |
| Taiping North Road | $PM_{2.5}$ | 0.031 | 0.479 ** | 0.357 * | −0.424 ** | 0.146 | −0.173 | 0.072 |
| | $PM_{10}$ | −0.172 | −0.603 ** | −0.238 | 0.256 | −0.162 | 0.187 | −0.024 |
| Jiyin Avenue | $PM_{2.5}$ | 0.348 ** | 0.437 ** | 0.156 | 0.220 | 0.311 * | −0.310 * | 0.412 ** |
| | $PM_{10}$ | −0.255 * | −0.422 ** | −0.053 | −0.111 | −0.347 ** | 0.339 ** | −0.413 ** |
| Shuanglong Avenue | $PM_{2.5}$ | −0.043 | 0.004 | −0.161 | −0.185 | 0.118 | −0.141 | 0.004 |
| | $PM_{10}$ | 0.013 | −0.036 | 0.228 | 0.253 * | −0.114 | 0.128 | −0.006 |
| Mengdu Street | $PM_{2.5}$ | −0.334 ** | −0.128 | 0.127 | 0.017 | 0.246 | −0.257 * | 0.297 * |
| | $PM_{10}$ | 0.268 * | 0.244 | 0.023 | −0.219 | −0.341 ** | 0.357 ** | −0.129 |
| Xianlin Avenue | $PM_{2.5}$ | 0.108 | 0.082 | −0.004 | −0.217 | 0.122 | −0.177 | −0.135 |
| | $PM_{10}$ | −0.145 | −0.092 | 0.115 | 0.204 | −0.063 | 0.132 | 0.124 |

** Indicates that the correlations was extremely significant. * Indicates that they had correlations.

### 3.2. Spatial Distribution of PM Concentrations among Plant Communities

#### 3.2.1. Ambient $PM_{10}$ and $PM_{2.5}$ Concentrations among Plant Communities at Different Distances from the Road

The ambient $PM_{10}$ and $PM_{2.5}$ concentrations at a height of 1.5 m but at different distances from Mufu South Road were used as an example (Figure 4a). The inhalable particles in Mufu South Road improved with the distance from the green space to the road, but the B1 plots on Mufu South Road fluctuated greatly because of the diversity of the plant communities. The inhalable PM in Mufu South Road increased to a certain extent with the appearance of the arbor, shrub, and grass communities, but the concentration decreased significantly in open spaces such as sidewalks and A1 plots. In general, the level of inhalable PM in the A1 lawn plot (control group) was slightly lower than that in the B1 tree–shrub–grass plant community group (Figure 4b).

#### 3.2.2. Ambient $PM_{10}$ and $PM_{2.5}$ Concentrations among Plant Communities at Different Heights above the Ground

We selected the A1 lawn communities and the B1 plot arbor, shrub, and grass communities along Mufu South Road for comparative analysis because there were fewer variables of the two research sites on the same road (Figure 5a). In plot B1, the concentration of $PM_{2.5}$ and $PM_{10}$ increased significantly at 0.5–1.5 m. The height of the shrubs (0–0.5 m) and the tree canopy (1.5–4 m) had a certain reduction effect on the pollutants. Since the height of trees is mostly below 4 m, no difference was found in the concentration of $PM_{2.5}$ and $PM_{10}$ above 4 m. However, in the A1 plot, the concentrations were lesser compared with those in the B1 plot in the range of 0–0.5 m, but the fluctuation of pollutant concentrations at

the heights of 1.5 m, 3 m, and 6 m was the same as that of the B1 plot, and the reduction effect was not significant. The experiments showed that the height of the leaves in the road greenbelt reduced the inhalable particulate matter. However, this had an accumulation and sedimentation effect on pollutants at the branch point under the tree canopy, where there were no leaves and only tree trunks, thereby increasing the concentration (Figure 5b).

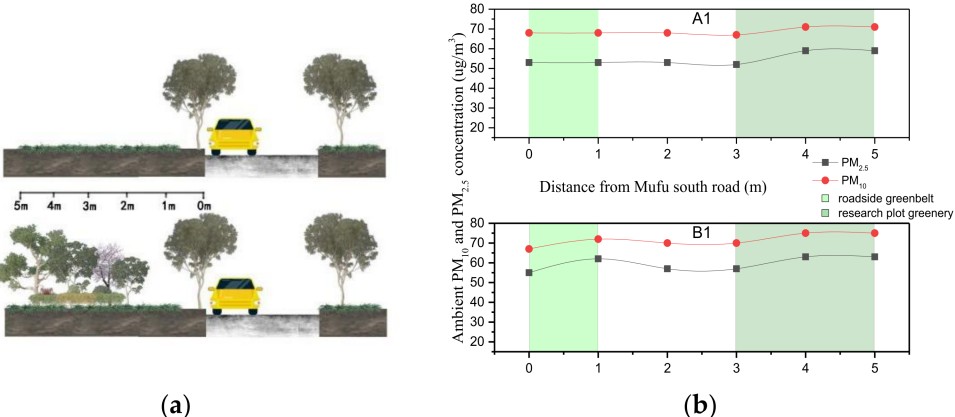

(a)                                                          (b)

**Figure 4.** (**a**) Elevation of A1 and B1 on Mufu South road. (**b**) Ambient PM$_{2.5}$ and PM$_{10}$ by distance from road.

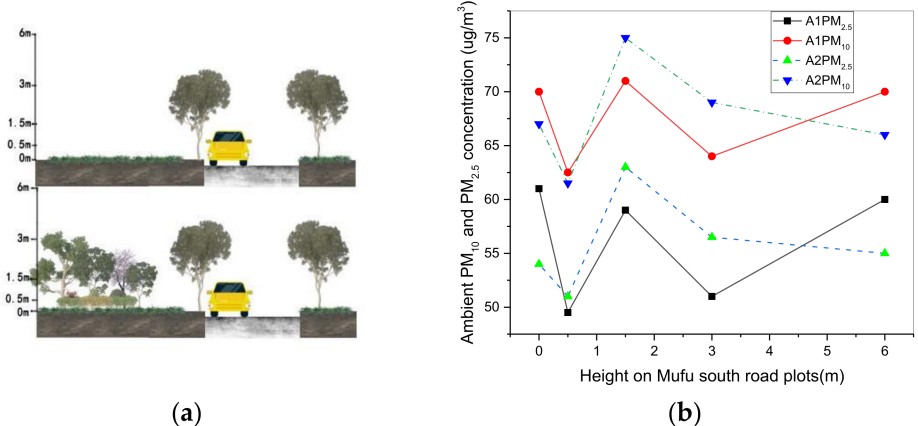

(a)                                                          (b)

**Figure 5.** (**a**) Elevation of A1 and B1 on Mufu South road. (**b**) Ambient PM$_{2.5}$ and PM$_{10}$ by height.

### 3.2.3. Different Ambient PM$_{10}$ and PM$_{2.5}$ Concentrations among Different Plant Community Structures

Six different plant communities were selected among the 14 research plots on the seven roads (square, lawn, arbor, arbor+shrub, arbor + grass, and arbor + shrub + grass). The comparison and analysis of the average reduction rate of different experimental plots showed that the rate of the reduction in PM$_{2.5}$ was as follows: arbor–shrub–grass > arbor–shrub > arbor–grass > arbor > grass > blank control. The rate of the reduction in PM$_{10}$ was as follows: tree–shrub > arbor–shrub–grass > arbor > arbor–grass > grassland > control group (Table 5).

The reduction rate fluctuations in the ambient PM$_{10}$ and PM$_{2.5}$ concentrations of the six plots were analyzed. The reduction rates were negative in the square and lawn plots, indicating that these two plots affected ambient PM concentrations (Table 5). In addition, the reduction rate of PM$_{10}$ in the arbor + shrub plot (19.11%) was greater than that of PM$_{2.5}$ (15.68%), but the reduction rate of PM$_{2.5}$ in the arbor + grass (15.19%) and arbor + shrub + grass (22.39%) plots was greater than that of PM$_{10}$ (9.41% and 14.29%, respectively). A correlation analysis between the reduction rates of ambient PM concentrations and

vegetation composition also showed positive correlations among the composition variables of plant communities (Figure 6).

**Table 5.** Reduction rates of different vegetation compositions.

|  | Plant Communities | $PM_{2.5}$ (ug/m³) | $PM_{10}$ (ug/m³) |
|---|---|---|---|
| Beijing East Road A3 (CK) | Square | −19.05% | −9.93% |
| Mufu South Road plot A1 | Lawn | −11.32% | −4.41% |
| Jiyin Avenue plot A7 | Arbor | 10.86% | 12.28% |
| Jiyin Avenue plot B7 | Arbor + shrub | 15.68% | 19.11% |
| Beijing East Road B3 | Arbor + grass | 15.19% | 9.41% |
| Taiping North Road plot B4 | Arbor + shrub + grass | 22.39% | 14.29% |

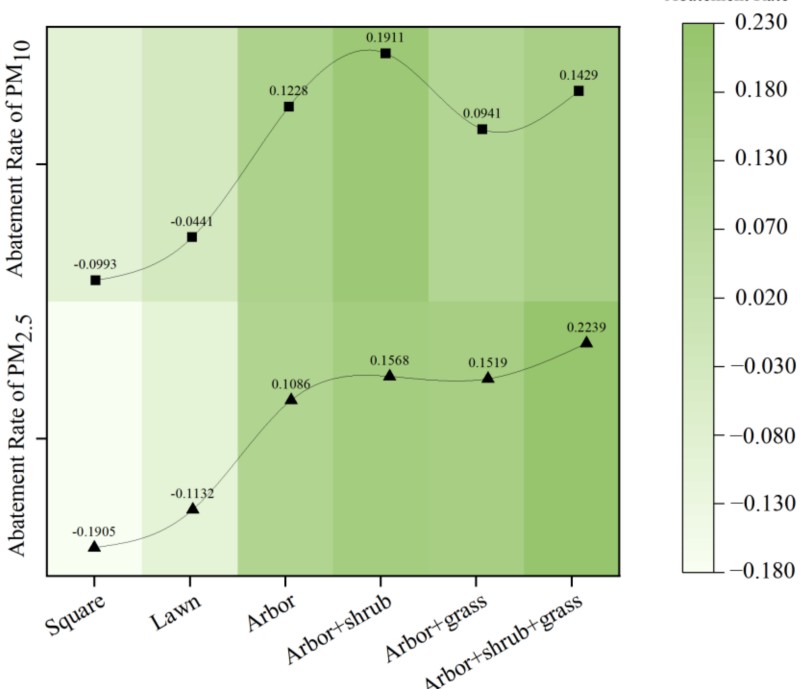

**Figure 6.** Reduction rates of different vegetation compositions. (Triangle represents ambatement rate of $PM_{2.5}$, Square represents ambatement rate of $PM_{10}$).

### 3.2.4. Different Ambient $PM_{10}$ and $PM_{2.5}$ Concentrations among Different Arbor Species

We selected three types of arbor species—coniferous, evergreen, and mixed broad-leaved—for comparison. We found that the reduction ability of the forest is in the following order: coniferous > deciduous > evergreen. The reduction ability of mixed forest is in the following order: mixed broadleaf + coniferous forest > mixed deciduous + coniferous forest > mixed evergreen + deciduous forest. In addition, neither pure evergreen arbor forest nor mixed evergreen + deciduous forest reduced the concentration of ambient $PM_{10}$ and $PM_{2.5}$ (Table 6).

Pure deciduous forests, mixed coniferous+broadleaf arbor, and mixed coniferous + deciduous forests have a stronger ability to reduce $PM_{2.5}$ than $PM_{10}$. In addition, pure coniferous forests had a stronger ability to reduce $PM_{10}$ than $PM_{2.5}$. The results show that mixed-tree communities have a greater impact on reducing $PM_{2.5}$ concentrations (Figure 7).

**Table 6.** Reduction rates of different arbor species.

|  | Arbor Species | $PM_{2.5}$ (ug/m$^3$) | $PM_{10}$ (ug/m$^3$) |
|---|---|---|---|
| Beijing East Road A3 (CK) | Square | −19.05% | −9.93% |
| Xianlin Avenue Road plot A2 | Conifer arbor | 12.90% | 16.16% |
| Shuanglong Avenue plot A6 | Deciduous arbor | 9.10% | 7.37% |
| Mufu South Road plot B1 | Evergreen arbor | −7.70% | −7.91% |
| Taiping North Road plot B4 | Broadleaf + conifer arbor | 22.39% | 14.29% |
| Shuanglong Avenue plot B6 | Deciduous + evergreen arbor | −3.65% | −5.68% |
| Taiping North Road plot A4 | Deciduous + conifer arbor | 15.38% | 7.33% |

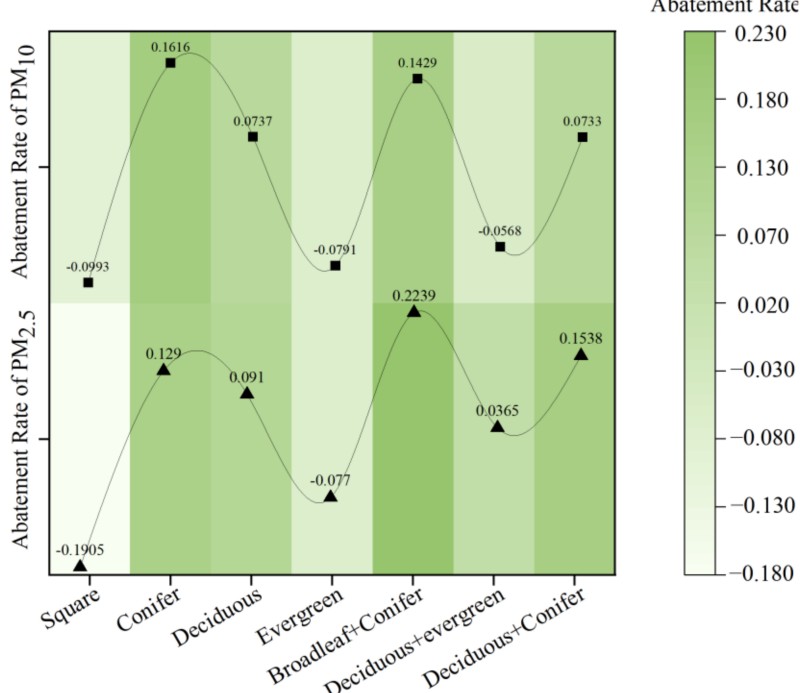

**Figure 7.** Reduction rates of different arbor species. (Triangle represents ambatement rate of $PM_{2.5}$, Square represents ambatement rate of $PM_{10}$).

## 4. Discussion

### 4.1. Effects of Vegetation Composition on PM Concentrations

Urban road greenbelts are considered a main factor in the reduction in ambient PM concentrations in cities [22–24]. Thus far, research has been primarily focused on the use of stationary equipment to monitor pollutants on road greenbelts to study the impact of vegetation on environmental pollutants. Plant communities and species characteristics have been regarded as key factors in reducing PM concentration levels [6,25,26].

The association between road plant community factors and pollutant concentrations is complex and not completely elucidated. Therefore, in this study, pollutant concentrations were detected in selected plant communities, and the reduction rates of different plant communities were calculated and analyzed. The results show that plant communities with trees, shrubs, and grasses have the best reduction effect on PM concentrations, followed by trees+shrubs, and then pure trees and trees + grass. Grass has little effect on the deposition of pollutants in plant groups, but shrubs and trees have a greater impact on the vegetation community. These results indicate that the larger a single plant is, the more PM it can reduce. Plant communities with mature or large plants (larger DBH) retained dust and reduced PM concentrations more effectively. Mori et al. [27], Liu et al. [28], and Li et al. [29] also reported similar results. Individual plant size affects the amount of ambient pollutant removal in plant communities [30]. The plant communities' reduction in PM concentrations

is associated with the increase in 3D green quantity. For example, Sheng et al. [9] found that the PM concentration index was positively correlated with 3D green quantity. Several studies have also demonstrated that plots with a larger canopy and multiple-layer structure reduce PM more effectively [30–32].

The layer structure of a plant community also affects the concentration of PM [33]. In this study, in the different plots on the same road, the plant communities with multi-layered structures reduced PM more efficiently, which concurred with the results of Jim and Chen et al. [30]. The monitoring time of this study was winter. The reduction in PM concentrations in multi-layered structural (arbor+shrub+grass) plant communities ($PM_{2.5}$, 22.39%; $PM_{10}$, 14.29%) was much higher than that in single-layer structural (pure arbor) plant communities ($PM_{2.5}$, 10.86%; $PM_{10}$, 12.28%). Lower PM concentrations were always found in the multi-layered structural plots (B3, B4, and B7), comprising *Styphnolobium japonicum + Prunus serrulata*; *Osmanthus fragrans + Phyllostachys edulis + Magnolia grandiflora + Zelkova serrata + Carya illinoinensis + Prunus serrulata*; *Koelreuteria paniculata + Osmanthus fragrans + Lagerstroemia indica + Prunus serrulata + Nerium oleander + Prunus persica*.

### 4.2. Arbor Species and Their Effects on PM Concentrations

Previous research has reported that grass and shrubs can reduce the concentration of ambient PM [34,35] by covering earth surfaces, holding ambient PM, and affecting meteorological factors [36,37]. The structure and species of plant communities, especially the characteristics of individual plant leaves, can significantly affect the PM concentrations in specific plant communities [21]. For instance, Baraldi et al. [38] showed that fluffy or sticky leaves can retain more dust or PM than leaves with smooth surfaces, and leaves with a greater leaf area index can significantly reduce ambient PM [39,40].

*Pinus*, *Cedrus deodara* (Roxb.) G. Don, *Metasequoia glyptostroboides*, Hu and W. C. Cheng, and *Taxodium distichum* var. *imbricatum* (Nuttall), Croom, have shown the largest effect on ambient PM concentrations. Most of these plant communities contain conifer trees. Therefore, conifers have demonstrated efficiency in adsorbing ambient PM [41,42].

The leaves of the plants can effectively remove ambient PM. Therefore, evergreen arbor species could reduce more ambient PM than deciduous arbor species during winter. However, the canopy density of evergreen arbor plants may influence the diffusion of pollutants [43]. Particular differences were found on the roads in Nanjing where PM concentrations were extremely high. In this research, regarding the reduction in the concentration of ambient PM by tree species in the plant community (conifer trees > deciduous trees > evergreen broad-leaved trees) and the result of the mixed forest abatement rate, coniferous+broad-leaved trees in mixed forests have the best reduction ability. The reduction rates of $PM_{10}$ and $PM_{2.5}$ were 14.29% and 22.39%, respectively.

### 4.3. Influence of Meteorological Factors on Pollutant Changes

Meteorological factors also have a certain impact on the concentration of airborne pollutants. In this study, air RH impacted ambient PM concentrations; however, the deposition velocities of PM and RH showed a positive correlation [7]. This result is similar to the findings of Tiwari et al. [44]. Previous reports also showed that no simple linear relations have been found between RH and ambient PM concentration; thus, the wetness of plant communities has complex effects on ambient PM concentration [45,46].

Air temperature affects the concentration level of pollutants around the plant community, and the reduction level of PM will increase with an increase in air temperature, consistent with the research of Xun et al. [47]. Additionally, the amount of light determines temperature change to a certain extent, consistent with the effect of temperature [48]. On the surface, the increased air temperature will increase the temperature difference between the inside and the outside of the plant canopy, generating gas flow and driving the diffusion of pollutants [49]. Therefore, the temperature difference and the distribution of gas pollutants are positively correlated, which plays a positive role in reducing pollutants in the plant community. In this study, we also found that the concentration of PM is highly

correlated with temperature and RH. The concentration of PM is negatively correlated with temperature, light, and R, but positively correlated with RH and traffic flow.

## 5. Conclusions

In this study, we present results on the effects of 14 urban roadside plant communities on ambient PM. The results showed that space openness and plant density affect ambient PM concentrations within plant communities. We compared the pollutant reduction rates of different plant community combinations and found that the ability of plant communities to reduce ambient $PM_{10}$ concentrations was related to their hierarchical spatial richness. The reduction rate of different arbor species was analyzed, and the key factors affecting the effect of arbor species on PM concentration were determined, among which conifers had the greatest impact on the reduction in PM concentration. In addition, the relative humidity and temperature in the plant community significantly affected the PM concentration. Changes in road traffic flow also affected the PM concentration to a certain extent. These findings will allow the prediction of the impact of road green space plant community structure and plant species on environmental PM, helping urban road planners and environmentalists mitigate PM associated with air pollution.

## 6. Limitations and Future Research

Our study focused on PM reduction using plant communities; however, we did not consider the reduction in other pollutants by such plant communities and the relationship between the concentrations of other pollutants and particulate matter. In future research, we can further study the relationship between urban green space and the six other pollutants.

**Author Contributions:** Conceptualization, C.L. and A.D.; methodology, C.L.; software, A.D.; validation, C.L. and A.D.; formal analysis, C.L.; investigation, C.L. and A.D.; resources, C.L.; data curation, Y.J.; writing—original draft preparation, C.L.; writing—review and editing, C.L.; visualization, A.D.; supervision, Q.S.; project administration, Z.Z.; funding acquisition, Z.Z. All authors have read and agreed to the published version of the manuscript.

**Funding:** This research was funded by the Natural Science Research of Jiangsu Higher Education Institutions of China (21KJB220008); Research results of Jiangsu Social Science Fund Project (21GLC002); Humanities and Social Sciences Research Project of the Ministry of Education "Research on the New Mechanism of Urban Green Space Ecological Benefit Measurement and High-quality Coordinated Development–Taking Nanjing Metropolitan Circle as an Example" (21YJCZH131); National Natural Science Foundation for Youth (32101582); Jiangsu Natural Science Foundation for Youth (BK20210613); and the APC was funded by (32101582).

**Institutional Review Board Statement:** Not applicable.

**Informed Consent Statement:** Not applicable.

**Data Availability Statement:** Not applicable.

**Conflicts of Interest:** The authors declare no conflict of interest.

## Abbreviations

Altitude (A); atmospheric pressure (AP); canopy area (CA); distance from the edge of the building (DB); diameter at breast height (DBH); distance from the road (DR); the direction of the wind (DW); the height of the tree (H); illumination (I); longitude and latitude (LO and LA); noise (N); particulate matter (PM); radiation (R); relative humidity (RH); temperature (T); wind speed (WS).

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
