# Peer review of "Effect of Different Plant Communities on Fine Particle Removal in an Urban Road Greenbelt and Its Key Factors in Nanjing, China"

_sustainability, doi:10.3390/su15010156_

Round 1
Reviewer 1 Report
Comments on manuscript titled “Effect of different plant communities on fine particle removal in an urban road greenbelt and its key factors in Nanjing, China” by Congzhe Liu, Anqi Dai,Yaou Ji, Qianqian Sheng, and Zunling Zhu
(Manuscript Number: sustainability-1942314)
The correlation between the distribution of plant communities and ambient PM concentrations in city road greenbelt has specific spatial patterns. The authors selected 14 plant community-monitoring sites on seven roads in Nanjing as research targets and monitored these roads in January for various parameters such as PM with aerodynamic diameters ≤10 μm (PM10) and PM with aerodynamic diameters ≤2.5 μm (PM2.5). The consequences revealed that the composition and species of plant communities effected the concentration of ambient PM directly, the ability of plant community structure to reduce ambient PM is in the order: trees+shrubs+grasses> trees+shrubs> trees+grasses> pure trees. Regarding the reduction of ambient PM by tree species in the plant community (conifer trees> deciduous trees> evergreen broad-leaved trees) and the result of the mixed forest abatement rate, coniferous+ broad-leaved trees in mixed forests have the best reduction ability, their reduction rates of PM10 and PM2.5 were 14.29% and 22.39%, respectively. The environmental climate indices of the road community, temperature, and traffic flow were positively correlated with ambient particulate matter, but relative humidity was negatively correlated with ambient particulate matter. Among them, PM2.5 and PM10 were significantly related to temperature and humidity, and the more open the green space on the road, the higher the correlation degree, and PM10 is also related to light and atmospheric radiation. These characteristics of plant community and the meteorological factors on urban road are the foundation of urban greenery ecological services, and the research showed that the adjustment of plant communities could improve greenbelt ecological services by reducing the concentration of ambient PM.
The conclusions of this manuscript are believable. I recommend that the manuscript be accepted for publication with minor revision in Sustainability.
Some suggestions:
1) The English should be improved.
2) The results should be explained in more detail. For example, 3.1. The influence of environmental factors on Ambient PM10 and PM2.5 concentration. The positive correlation of the concentrations of ambient PM10 & PM2.5 and traffic flow are not expressed in detail. The differences of the positive correlations in the morning, at noon and at evening, as well as the cause should be explained clearly.
3) By analyzing the partial correlations of various meteorological factors and road pollutants on the seven roads under winter conditions, the authors found that the concentration of road pollutants is affected by complex meteorological factors. However, the standard of partial correlations of various meteorological factors and road pollutants should be clearly expressed and explained in detail.
Reviewer 2 Report
Dear editor and authors,
Thank you for giving me an opportunity to review the paper entitled: ,, Effect of different plant communities on fine particle in an urban road greenbelt and its key factors in Nanjing, China,,.
The introduction section should be included as follows
1. Why such study with proposed research gaps is important?
2. How this research gap relates to current issue?
3.
2.5 Data analysis. What are the advantages and disadvantages of the statistical analysis used?
4. Explain in more detail the standard of partial correlations of different meteorological factors and road pollutants.
5. Limitations and future research
Best regards
